# Are Faba Bean and Pea Proteins Potential Whey Protein Substitutes in Infant Formulas? An In Vitro Dynamic Digestion Approach

**DOI:** 10.3390/foods9030362

**Published:** 2020-03-20

**Authors:** Linda Le Roux, Olivia Ménard, Raphaël Chacon, Didier Dupont, Romain Jeantet, Amélie Deglaire, Françoise Nau

**Affiliations:** 1Sill Dairy International, Raden, 29860 Plouvien, France; linda.leroux@sill.fr (L.L.R.); raphael.chacon@sill.fr (R.C.); 2STLO, INRAE, AGROCAMPUS OUEST, 35042 Rennes, France; olivia.menard@inrae.fr (O.M.); didier.dupont@inrae.fr (D.D.); romain.jeantet@agrocampus-ouest.fr (R.J.); amelie.deglaire@agrocampus-ouest.fr (A.D.)

**Keywords:** infant formula, plant protein, in vitro digestion, microstructure, proteolysis, apparent protein digestibility

## Abstract

Infant formulas (IFs) are used as substitutes for human milk and are mostly based on cow milk proteins. For sustainability reasons, animal protein alternatives in food are increasingly being considered, as plant proteins offer interesting nutritional and functional benefits for the development of innovative IFs. This study aimed to assess how a partial substitution (50%) of dairy proteins with faba bean and pea proteins influenced the digestibility of IFs under simulated dynamic in vitro digestion, which were set up to mimic infant digestion. Pea- and faba bean-based IFs (PIF and FIF, respectively) have led to a faster aggregation than the reference milk-based IF (RIF) in the gastric compartment; that did not affect the digesta microstructure at the end of digestion. The extent of proteolysis was estimated via the hydrolysis degree, which was the highest for FIF (73%) and the lowest for RIF (50%). Finally, it was apparent that in vitro protein digestibility and protein digestibility-corrected amino acid score (PDCAAS)-like scores were similar for RIF and FIF (90% digestibility; 75% PDCAAS), *but* lower for PIF (75%; 67%). Therefore, this study confirms that faba bean proteins could be a good candidate for partial substitution of whey proteins in IFs from a nutritional point of view, provided that these in vitro results are confirmed in vivo.

## 1. Introduction

Infant formulas (IFs) are primarily marketed as powders and used as substitutes for human milk; IFs are made to mimic the nutritional composition of human milk as much as possible. IF powders are made using spray drying technology, which extends their shelf-life and aids in handling, especially when compared to liquid IFs [1]. Their macronutrient (carbohydrates, fat, and proteins) and micronutrient (minerals and vitamins) content is defined to cover the nutritional requirements for infants, with particular attention given to protein intake, which is essential early in life [2]. The average infant protein requirement ranges from 1.1 g/kg/day at age 6 months to 0.7 g/kg/day at 10 years, before a small decline towards the adult value thereafter [3]. According to the applicable European regulation, the sources of protein allowed for first age IFs (0 to 6 months) are either cow milk’s protein, goat’s milk protein, soy protein isolate, or hydrolyzed rice protein [4]. However, in this last regulation, it was mentioned that other ingredients could be used in IFs if their safety and nutritional balance were in accordance with infant requirements. In this way, this study aims to investigate new possibilities in the field of innovation in IFs by studying the possibility of a partial substitution of cow milk proteins via alternative plant proteins not already used in IFs.

In fact, an explosion in the demand for animal proteins is expected because of population growth and an increased standard of living in developing countries, which is projected to double by 2050 [5,6]. Therefore, alternative protein sources that show a similar nutritional value to animal proteins are now explored. It is in this way that plant proteins are seen more as potential replacers of animal proteins in food [7], especially in legume proteins such as soy, pea, chickpea, faba beans, or lupine proteins. All of these sources are becoming a good alternative to animal proteins [7,8,9] because of their high nutritive quality and good techno-functionalities including solubility and emulsifying properties [10,11]. While soy proteins remain the most consumed plant proteins, pea proteins (*Pisum sativum*) are becoming another viable alternative to animal proteins because of techno-functional and nutritive characteristics. Pea protein present a high essential amino acid (EAA) content [12] and relatively good digestibility [13], very similar to soybean proteins [14,15]. Furthermore, pea seeds have a lower content of anti-nutritive components, such as proteinase inhibitors and phytic acid [16], which cause less frequent allergic reactions in children than other legumes [17]. Faba beans (*Vicia faba L.*) are also a good source of quality protein, as they are particularly rich in lysine and threonine. Its use in food products as a potential soy substitute has been promoted, even more so as the cultivation of faba bean has beneficial effects as part of sustainable agroecosystems [18]. In their raw state, faba bean proteins also contain anti-nutritional factors, such as tannins, vicine, and convicine compounds that can reduce protein digestibility and lead to pathologic conditions [18,19]. However, there has been considerable progress through research and development to improve both the nutritional and functional properties of plant proteins. For instance, specific technological or chemical treatments enable to inactivate or inhibit most of the anti-nutritional factors [20,21,22] and thus improve the biological value and the digestibility of plant proteins. Thus, the question arises whether pea and faba bean proteins are conceivable for first age IFs as alternative plant proteins to soy or hydrolyzed rice proteins that are already used in IFs.

The overall concept of protein digestibility is relatively simple. It is defined as the ratio of the difference between the ingested and excreted nitrogen to the ingested nitrogen. However, the in vivo measurement is actually complicated. For that reason, several criteria resulting from in vitro approaches have been suggested to approximate protein digestibility that can be suitable to predict protein and amino acid (AA) utilization by the human body [23,24,25]. The in vitro protein digestion has been studied and calculated using different methods to compare the digestibility of human milk, cow milk, as well as cow milk-, goat milk-, or soy protein-based IFs: proteolysis measurements using SDS-PAGE (Sodium dodecyl sulphate polyacrylamide gel electrophoresis) coupled with density methods [26,27,28], the pH drop method [27] or a more accurate method that uses a ratio of bioaccessible AA or nitrogen content in dialysate samples after digestion compared to the AA or nitrogen intake [29]. Moreover, the ability to use plant proteins in IFs has also been investigated, but most of the studies concerned the use of chickpea protein in follow-on formulas (6 to 12 months) [30,31]. Some others have focused on the capacity of pea protein [32] or different other legume proteins (chickpea, faba bean, lentil, and pea) [33] for probiotics encapsulation in follow-on IFs. Recently, a process for preparing a first age IF-based potato protein has been patented [WO2018 115, 340 (A1)]. All these relevant studies explore the possibility of plant protein use in IFs, proving that they deserve to be further considered along with other protein sources as they could be suitable for infant needs from birth.

In a previous study [34], four plant proteins were selected (EAA profile adapted to infants, no allergen, no organoleptic defects, commercially available, innovative source) for replacing the whey protein concentrate usually added to skimmed cow milk. The results corresponded to 50% of the total protein in IF. IFs have been prepared at a pilot scale and submitted to in vitro static digestion. While rice and potato IFs showed limitations in terms of manufacturing and digestibility, pea and faba bean IFs (PIF and FIF, respectively) showed functional properties and overall digestibility closer to the milk-reference IF (RIF).

The purpose of the present study was to investigate further the aptitude of these two plant proteins (pea and faba bean) to substitute whey proteins in IFs produced at a semi-industrial scale [35], by testing them using an in vitro digestion model in dynamic conditions. The dynamic in vitro digestion model was set up to mimic infant digestion on the basis of an extensive analysis of literature on infant physiology [36]. The configuration has been previously validated using in vivo data [37]. During digestion, the microstructure of the digestas was studied and several indicators were followed-up, such as the degree of protein hydrolysis (DH), the in vitro protein digestibility, and the bioaccessibility of EAA. An adaptation of the protein quality measurement developed by the Food and Agriculture Organization (FAO) [38], the protein digestibility-corrected amino acid score (PDCAAS), was calculated in which the protein digestibility usually determined in vivo was replaced by the in vitro protein digestibility.

To the best of our knowledge, this is the first time that first age IFs (0 to 6 months) containing plant proteins other than soy and hydrolyzed rice proteins have been reported, designed, and submitted to dynamic in vitro digestion.

## 2. Materials and Methods

### 2.1. Chemicals

Porcine pepsin (P7012; 2971 IU/mg), porcine pancreatin (P7545; 6.79 IU/mg), bovine bile extract (B8631; 3.1 mmol/g), and the enzyme inhibitors pepstatin A (P5318) and pefabloc (76,307) were all obtained from Sigma-Aldrich (St. Quentin Fallavier, France). Bovine bile salts concentration and enzyme activities were determined as described in the Electronic Supplementary Information of [39]. The fluorescent dyes used for Confocal laser scanning microscopy (CLSM) analysis were high-content screening (HCS) LipidTOX™ Green Neutral Lipid Stain (H34475) obtained from Thermo Fisher Scientific (Illkirch, France), Rd-DOPE® Liss Rhod PE 1,2-dipalmitoyl-sn-glycero-3-phosphoethanolamine-N-(lissamine rhodamine B sulfonyl) (810150C), and Fast Green Food Green 3 (FCF) (Sigma F7258, Sigma-Aldrich, St Louis, MO, USA). The standards used for size exclusion chromatography were purchased from Phenomenex (Waters Inc., Milford, MA, USA) (No. ALO-3042 for bovine thyroglobulin, IgA, IgG, ovalbumin and myoglobin) and from Sigma-Aldrich (cytochrome C (C2506) and human angiotensin II (A9525)). All other chemicals were of standard analytical grade.

### 2.2. Infant Formulas (IFs)

A reference IF (RIF, containing cow milk proteins only) and two IFs containing 50% milk proteins and 50% either pea proteins (PIF) or faba bean proteins (FIF), were manufactured using processing conditions as described in Le Roux et al. (2020) [35]. The three IF powders were suspended in 150 mL water and preheated at 37 °C in a baby bottle to reach a protein content of 1.3%. The nutritional composition of the IFs is presented in Section 3.1. The protein content of the reconstituted IFs was determined using the Kjeldhal method [40] and a conversion nitrogen factor of 6.38 for RIF [41] and 5.9 for PIF and FIF. The latter factor is the average of the one for bovine milk proteins (6.38) and the one for pea or faba bean proteins (5.4 of which corresponded to an average for legume proteins, ranging from 5.24–6.64 [41]). The EAA content was determined as described in Section 2.4.3.

### 2.3. In Vitro Dynamic Digestion

Gastrointestinal digestions of RIF (reference IF), PIF (pea IF), and FIF (faba bean IF) were performed in an in vitro dynamic system (DIDGI^®^, INRA, Paris, France). Parameters for gastric and intestinal phases were chosen to closely mimic the digestive conditions of term newborn fed with human milk at the postnatal age of four weeks and have been adapted from de Oliveira et al. (2016) [42] (Table 1). The main change was the gastric emptying half-time (t ½) of 78 min for an IF [37]. The in vitro dynamic system was controlled by the STORM^®^ software, which allows regulating and monitoring the digestive parameters. No gastric lipase was used since the objective of the present study was focused on protein digestion.

Digestion experiments were performed over three hours and in triplicate for each IF. Samples were collected from the IFs before digestion (time 0 min) and in both compartments at 30, 60, 90, 120, and 180 min after the beginning of the digestion. Digesta emptied from the intestinal compartment was collected on ice over the three hours of digestion. Structural analyses was conducted using confocal microscopy. Laser light scattering was performed immediately after sampling (see Section 2.4.1). Protease inhibitors, namely 10 µL of 0.72 mM Pepstatin A per mL gastric digesta or 50 µL of 0.1 M Pefabloc per mL intestinal digesta, were immediately added to the samples taken for proteolysis analysis, before freezing at −20 °C. Digestion was performed in triplicate for each IF.

### 2.4. Digesta Analysis

#### 2.4.1. Structural Characterization 

The microstructure of the IFs and of the digestas was observed using two methods, as previously described by Bourlieu et al. (2015) [45]. Confocal laser scanning microscopy (CLSM) was performed using a Nikon C1Si inverted microscope TE2000-E (Nikon, Champigny-sur-Marne, France) equipped with a × 100 oil immersion objective; three fluorescent dyes were added to the samples (Lipidtox™ for apolar lipids, Rhodamine-PE for amphiphilic compounds, and Fast Green FCF for proteins). At least 5 to 10 images were taken of each sample. Laser light scattering was completed using a Mastersizer 2000 (Malvern Instruments, Malvern, UK), which was equipped with two laser sources for analyzing the particle size distribution. The diameter modes (i.e., the particle diameters of the most frequent particles in the volume distribution) were also measured.

#### 2.4.2. Degree of Hydrolysis (DH)

The DH was calculated from the measurement of the primary amines released during digestion. Primary amines were measured in the soluble fraction of the samples, obtained after centrifugation for 20 min at 10,000·*g*, and at 4 °C, using the o-phthaldialdehyde (OPA) method according to Darrouzet-Nardi et al. (2013) [46]. The absorbance was measured at 340 nm with a Multiskan™ GO Microplate Spectrophotometer (Thermo Fisher Scientific, Waltham, MA, USA). A calibration curve was prepared using methionine standard solutions (0 to 2 mM). The DH was calculated as follows:(1)$$\%\;\mathrm{DH}=100\;\times \;\left[\frac{\mathrm{NH}2\left(t\right)-\mathrm{NH}2\left(t0\right)}{\mathrm{NH}2\left(t0t\right)-\mathrm{NH}2\left(t0\right)}\right]$$
where *NH_2(t)_* was the concentration of primary amines after t min digestion, *NH_2(t0)_* was the concentration of primary amines in the IF before digestion, and *NH_2(tot)_* was the concentration of the total primary amines measured after total acid hydrolysis (HCl 6 N, 110 °C, 24 h) of the IF. All values were expressed as g per 100 g IF. All measurements were carried out in duplicate for each digesta.

#### 2.4.3. Amino Acid Analysis

The total AA contents were determined after acid hydrolysis of each IF, according to Davies and Thomas (1973) [47]. Acid hydrolysis of IF powder (20 mg) was performed by adding 2 mL of 6 N hydrochloric acid and heating at 110 °C for 24 h in vacuum sealed glass tubes. The sulfur-containing AA, cysteine, and methionine, were measured as methionine sulphone and cysteic acid after performic acid oxidation. The determination of tryptophan was not possible due to its degradation following acid hydrolysis. Total AA content of each IF was determined in duplicate.

The free AA contents were determined after deproteinization of the samples according to the method presented by Mondino et al. (1972) [48]. To this end, sulfosalicylic acid was added to digesta (0.05 g/mL), followed by incubation for 1 h at 4 °C and then centrifugation at 5000·*g* for 15 min at 4 °C. The supernatants were filtered through a 0.45 μm pore-size membrane (Sartorius, Palaiseau, France) and diluted five times with a 0.2 mol/L lithium citrate buffer (pH 2.2) before injection. Free AA content was determined once for each digestion experiment, i.e., in triplicate for each IF.

The AA analysis was carried out with cation exchange chromatography on a Biochrom30 automatic AA Analyser (Biochrom Ltd., Cambridge, G.B.), which was equipped with a cation exchange column 200 mm × 4.6 mm with a sulfonated polystyrene resin. Further, it was rreticulated via divinylbenzene and conditioned in lithium form, from Biochrom 30 (Serlabo technologies, Trappes, France). Samples were eluted with a 0.2 M lithium citrate buffer, pH 2.2, at 0.42 mL/min with post-column derivatization with ninhydrine (Ultra Ninhydrin Reagent Kit, Biochrom) according to Moore et al. (1958) [49]. The quantity of AA released during digestion was expressed as the percentage of free AA (expressed in g/100 g IF) related to the total AA (g/100 g IF).

#### 2.4.4. Soluble Nitrogen Content and Molecular Weight Distribution

IFs and intestinal digesta in the intestinal compartment at 3 h of digestion (or emptied from the intestinal compartment over 3 h) were analyzed for total N and soluble N (micro-Kjeldahl method) after the removal of insoluble particles by a 20 min centrifugation at 10,000·*g* and 4 °C. Molecular weight distributions of the resulting soluble fractions were determined by size exclusion chromatography (SEC), using a Biosep-SEC-2000 Phenomenex column connected to a Waters e2695 separation module equipped with a Waters e2489 UV/Visible detector (Waters Inc., Milford, MA, USA). Samples were eluted at 40 °C under isocratic 0.8 mL/min flow of 50 mM phosphate buffer pH 7 containing 0.2 M NaCl. Detection was performed at 214 nm. The column was previously calibrated by injecting eight molecular weight markers: bovine thyroglobulin (670 kDa), IgA (300 kDa), IgG (150 kDa) from human gamma globulin, ovalbumin (44 kDa), myoglobin (17 kDa), cytochrome C from horse heart (12 kDa), and human angiotensin II (1.05 kDa). The calibration curve (Log molecular weight versus retention time) allowed to determine the retention times defining the limits of each molecular weight range: >10 kDa, 10–5 kDa, 5–2 kDa, 2–1 kDa, and 1–0.2 kDa. The proportion of soluble proteins and peptides in a given molecular weight range was determined as the percentage of area under the curve between the respective limits (% Area SEC). Analysis was carried out in duplicate for each sample.

The soluble N fraction corresponds to the nitrogen contained in the proteins, peptides, and free AA of the soluble fraction. For each molecular size range, and because free amino acids are supposed to be undetectable at 214 nm, the proportion of soluble N in this range (% N SEC) was calculated as follows:(2)$$\%\;N\;\mathrm{SEC}\;(\times \;\mathrm{kDa})=\left[\frac{\mathrm{Total}\;\mathrm{soluble}\;N\;(\mathrm{Digesta})\;-\;\mathrm{Soluble}\;N\;(\mathrm{Free}\;\mathrm{AA})}{\mathrm{Total}\;N\;(\mathrm{IF})}\right]\;\times \;\%\;\mathrm{Area}\;\mathrm{SEC}\;(\times \;\mathrm{kDa}).\;$$

Total soluble *N _(Digesta)_* corresponded to the quantity of soluble *N* (mg) in the digesta at the end of digestion in the intestinal part (both intestinal compartment and intestinal emptied fraction). Total *N _(IF)_* corresponded to the total *N* in the IF (mg). Soluble *N _(Free AA)_* was calculated as the quantity of soluble *N* corresponding to free amino acids (free AA) (mg).

#### 2.4.5. In Vitro Digestibility and PDCAAS-Like Score

In vitro apparent protein digestibility was determined based on the soluble N lower than 10 kDa, i.e., as measured in the peptides by SEC and cumulated to the free AA nitrogen. It was determined in the intestinal compartment at 180 min and in the intestinal fraction emptied over 180 min. In both cases, it was calculated as follows:(3)$$\mathrm{In}\;\mathrm{vitro}\;\mathrm{Apparent}\;\mathrm{Protein}\;\mathrm{Digestibility}\;(\%)=\frac{\;\left(Ʃ\left[\;N\;\mathrm{SEC}\;\left(<10\;\mathrm{kDa}\right)\right]+\;\mathrm{Soluble}\;N\;\left(\mathrm{Free}\;\mathrm{AA}\right)\right)}{(N_{\mathrm{IF}}+{\;N}_{\mathrm{Sec}\mathrm{retions}\;}){\;x\;\%\mathrm{substrate}\;}_{\;}}\times 100.$$

$N_{SEC}\;$and *Soluble N _(Free AA)_* (expressed as mg/kg digesta) was the *soluble N* content in the intestinal compartment or in the intestinal emptied fraction; % substrate was the percentage of the IF initially introduced in the digester that was present in the intestinal compartment or in the intestinal emptied fraction (g IF/100 g digesta), estimated using the emptying equation; the known flows in the digester. $N_{IF}$ (expressed as mg/kg IF) was the total N content of the meal introduced in the digester. $N_{Secretions\;}$(expressed as mg/kg IF) was the total nitrogen content of the simulated bile secretion and pancreatin solution. Both digestibility values (intestinal compartment and emptied fraction) were averaged after weighting each value according to the substrate repartition in these two fractions.

The PDCAAS-like score (protein digestibility-corrected amino acid score) was calculated by adapting the methodology of FAO and WHO [39] in which the true fecal protein digestibility (normally determined in vivo in growing rats) was replaced by the in vitro apparent protein digestibility calculated as described above. The equation was then as follows:PDCAAS-like = Amino acid score (of the limiting AA) × In vitro Apparent Protein Digestibility (4)
(5)$$\mathrm{Amino}\;\mathrm{acid}\;\mathrm{score}\;(\mathrm{AAS})=\frac{\mathrm{AA}\;\mathrm{content}\;\mathrm{of}\;\mathrm{the}\;\mathrm{infant}\;\mathrm{formula}}{\mathrm{AA}\;\mathrm{content}\;\mathrm{of}\;\mathrm{the}\;\mathrm{reference}\;\mathrm{patter}}$$

As the present study focused on infants, the reference AA pattern used was that recommended for 0 to 6 months infants [50].

### 2.5. Statistical Analysis

Statistical analyses were conducted with the use of R version 3.5.2 [51]. The residues of the linear model with meal and digestion time (and their interaction) as factors were not normal for the kinetics of hydrolysis (DH) and the kinetics of EAA release, using the Kolmogorov–Smirnov test (“lillie.test” from the “nortest”package) [52]. Therefore, a non-parametric analysis for repeated measurements was completed for these two variables, taking the type of meal and the digestion time (and their interaction) into account, using the “f1.ld.f1” function of the package “nparLD” [53]. In the event of a significant treatment effect, the function “npar.t.test” or “nparcomp” of the R package “nparcomp” [54] was systematically used. In the event of a significant interaction effect, a linear mixed effect model with a random intercept on experiments to take into account the repeated measurements was performed and followed by the “difflsmeans” of the “lmerTest” package [55].

Regarding the nitrogen size distribution and the in vitro apparent digestibility and the PDCAAS-like score, a one-way ANOVA (“anova.lme” function from the “nlme” package) was completed with meal as the factor after verifying the residues of this model were normal with the Kolmogorov–Smirnov test (“lillie.test” from the “nortest” package) [52]. A post-hoc test (“LSD.test” of the “agricolae” package) was conducted when the differences were significant (*p* < 0.05). All results were expressed as mean ± SD.

## 3. Results and Discussion

This study aimed to explore the possibility of substituting whey proteins usually added to skimmed cow milk to formulate first age IFs with alternative plant-based proteins, wherein they would represent 50% of the total protein content. Specifically, we examined pea and faba bean proteins, as they have been considered worthy of further investigation due to their EAA profile, which is compatible with the nutritional requirements of the infants. Moreover, we investigates these proteins because their behavior was shown to be suitable with processing constraints [34]. Therefore, one reference IF (dairy-based) and two innovative plant-based IFs (pea and faba bean, respectively) have been produced at a semi-industrial scale [35] and submitted to an in vitro dynamic model of digestion in order to compare some relevant indicators of their protein nutritional quality.

### 3.1. Composition and Essential Amino Acid (EAA) Content of Infant Formulas

In order to assess the only effect of the protein source, the three IFs were designed equivalent in terms of calorie, protein, fat, and carbohydrates contents (Table 2). The EAA contents were not significantly different between the two plant-based IFs, except for methionine, phenylalanine, tyrosine, and lysine. The contents were lower in FIF compared to PIF, but the difference was less than 7%. However, contents in EAA—except for tyrosine, phenylalanine and histidine—were significantly lower in the two plant-based IFs compared to the reference RIF. The higher content of non-essential AAs in pea and faba bean proteins (10% higher) compared to milk proteins (Appendix A) can explain the lower total content of EAAs in plant-based IFs that are observed (Table 2).

### 3.2. Structural Changes in Gastric and Intestinal Compartments During Digestion of Infant Formulas

#### 3.2.1. Gastric Compartment

Before digestion (0 min), homogeneous and small particles of proteins and fat droplets were observed for RIF with unimodal distribution. Bimodal distributions were observed for PIF (pea infant formula) and FIF (faba bean infant formula). In these latter IFs, small fat droplets and protein particles coexisted with larger protein particles, especially in PIF (Figure 1A). The modal diameters before digestion were 0.6 µm for RIF, 0.8 and 10.0 µm for FIF, and 0.8 and 56.4 µm for PIF (Table 3). After 60 min of digestion, a majority of large aggregated particles of proteins and lipids were observed in the gastric compartment for PIF (31.7 µm modal diameter) and FIF (12.6 µm), whereas no large variations were seen for RIF (0.6 µm). After 120 min of digestion, only large aggregated particles were observed for all IFs, wherein proteins and lipids co-located (modes of 17.8 µm for RIF, 44.8 µm for PIF, 14.2 µm for FIF). FIF, however, had a small proportion of much smaller particles (0.1–0.2 µm) remain (Figure 1A).

The bigger particles found in PIF and FIF before digestion (0 min) suggest an incomplete solubilization of plant proteins, but protein aggregation during the technological processes might also occur. Indeed, heating of globular proteins above their denaturation temperature [56,57,58] leads to their unfolding, exposure of hydrophobic patches, and irreversible aggregation by forming hydrophobic interactions, hydrogen bonds, and/or disulfide bonds. Protein aggregation may depend on the protein nature and the physicochemical conditions (pH versus isoelectric point, nature, and concentration of salts, etc.). In any case, it influences the protein solubility [59,60,61]. After 60 min of digestion, the smallest particles initially present in the gastric compartment for PIF and FIF almost totally disappeared and were replaced by larger aggregated particles, which was likely due to acidification [45,62]. After 120 min of digestion, the microstructure of gastric digesta for PIF and FIF only slightly differ compared to that at 60 min of digestion, while for RIF, a strong aggregation occurred resulting in only large particles (mode of 17.8 µm), whereas the initial small particles completely disappeared.

From this, it is clear that the aggregation was delayed for RIF compared to PIF and FIF, which was most likely due to the protein source specificity. At 120 min of digestion, pH 4.9 was reached in the gastric compartment, which is close to the isoelectric point of the major cow milk caseins, i.e., β-casein and αs-casein (pI 4.2–5.1), and consequently induced casein aggregation in RIF. Since the net charge of the main whey proteins-i.e., β-lactoglobulin and α-lactalbumin (pI 5.1–5.2)—also decreased in these pH conditions, aggregation and/or interaction between whey proteins and uncharged lipids might also be involved [63,64,65]. However, the major pea proteins named legumin (Uniprot: P15838) had much higher isoelectric points, with a theoretical pI of 6.11 (calculated from the amino acid sequence of the protein using the Compute pI/MW tool of Expasy). Similarly, the legumin (Uniprot: P05190) from faba bean has a theoretical pI of 5.78. Thus, the presence in PIF and FIF of proteins with isoelectric points higher than that of whey proteins might explain the earlier aggregation in these infant IFs during the gastric phase, namely at 1 h digestion, when pH reached 5.9. Then, casein aggregation likely occurred in addition and similarly for all three IFs during the end of the gastric phase. Moreover, it has been described in previous work [66] that albumins from pea proteins (the second major proteins in pea) are resistant to pepsin action at pH 4.0. This may explain why bigger particles remain in solution at the end of the gastric digestion (120 min) for PIF compared to RIF and FIF (Figure 1A).

#### 3.2.2. Intestinal Compartment

At 60 min of digestion, the largest aggregates observed for PIF and FIF at the same time in the gastric compartment had largely disappeared in the intestinal compartment (Figure 1B). As observed by Ménard et al. (2014) [38], when arriving into the intestine, proteins are instantaneously hydrolyzed. This is explained by a higher solubilization of the proteins due to the neutralization of the pH and the large excess of digestive proteases [38]. This very likely reflects the intestinal hydrolysis, mostly responsible for the amphiphilic compounds observed on the CLSM images (Figure 1B) and corresponding to various products of digestion (peptides, fatty acids, etc.). Consequently, the modes at 60 min of digestion were comparable between the three IFs (modes for RIF: 0.2, 1.1 and 5.6 µm; PIF: 0.3, 1.2 and 15.8 µm; FIF: 0.6 and 11.2 µm). Even more, the distributions of particle sizes were nearly the same for RIF and PIF. Similarly, after 120 min of digestion, the particle size distributions in the intestinal compartment were rather analogous among the three IFs (modes of RIF: 0.2, 0.6 and 15.8 µm; PIF: 0.2, 0.8 and 17.8 µm; FIF: 0.6, 1.3 and 15.8 µm), especially for RIF and PIF.

To summarize, structural differences were mainly observed in the gastric compartment with different aggregation rates and protein particle size between the plant-protein based IFs and the cow’s milk protein IF. These physical differences occur during the gastric phase and may affect the rate of gastric emptying in vivo [67,68,69] and/or in non-homogeneous emptying from the stomach to the intestine. This, in turn, could have an effect on the absorption of nutrients in the upper part of the intestine and global nutrient metabolism in infants. Unfortunately, these phenomena could not be considered in our in vitro model, as the same rate of gastric emptying was applied for RIF, PIF, and FIF due to a lack of in vivo data.

### 3.3. Kinetics of Proteolysis

The kinetics of proteolysis were determined from the changes in the degree of hydrolysis (DH), which is defined as the proportion of cleaved peptide bonds [70]. In the gastric compartment, the IF effect was not significant (*p* > 0.05) but there were significant effects of time (*p* < 0.001) and of IF × time (*p* < 0.01). The DH significantly increased from 30 to 180 min, but only to a limited extent, as indicated by the DH reaching 180 min of digestion: 4.2 ± 1.0%, 8.2 ± 2.0% and 10.7 ± 0.7% for RIF, PIF, and FIF, respectively (Figure 2A). These low values of proteolysis can be explained by the reduced pepsin to protein ratio used, coupled with a relatively high gastric pH (pH between 4 and 6, while optimal pH for pepsin activity is closer to 2), both chosen to mimic an infant’s stomach conditions. Nevertheless, these DH values were higher than those measured (< 2%) at the end of the gastric phase when in vitro digestion was performed in static conditions [34]. Above all, the three IFs significantly differed with FIF DH higher than PIF DH and both higher than RIF DH. These results highlight differences between the three IFs during the dynamic in vitro gastric digestion that were not perceptible in static conditions and indicate a higher sensitivity to gastric conditions for pea and faba bean proteins compared to whey proteins.

As soon as the chyme entered the intestinal bowl, proteolysis drastically increased for all IFs, to reach 35.9 ± 2.5%, 44.3 ± 2.3%, and 49.7 ± 4.4% DH after 30 min of digestion for RIF, PIF, and FIF, respectively (Figure 2B), after which DH remained constant in this compartment up to 90 min of digestion (p > 0.05). Then, DH slightly increased in the intestinal bowl to finally reach 49.7 ± 4.2%, 66.3 ± 4.0%, and 72.8 ± 5.4% at 180 min of digestion for RIF, PIF, and FIF, respectively. Besides the expected significant effect of time (*p* < 0.001), IF also significantly impacts DH in the intestinal compartment (*p* < 0.001). In fact, the DH measured in the intestinal compartment significantly differed for the three IFs during the entire digestion (except at 90 min digestion), with DH values systematically higher for FIF than for PIF and RIF. This explains why the IF × time effect was not significant in the intestinal compartment (*p* > 0.05). These differences could be explained by the respective sensitivity of each protein to the intestinal enzymes, which was linked to the specificity of these enzymes. The porcine pancreatic enzymes used in the present study contained trypsin, which specifically cleaves carboxyl bonds after basic AA (lysine and arginine); chymotrypsin, which cleaves carboxyl bonds after aromatic AA (phenylalanine, tyrosine and tryptophan); and carboxypeptidases A and B, which cleave carboxyl bonds after aromatic AA and basic AA, respectively. Faba bean and pea protein isolates were found to contain more aromatic AA and arginine, whereas cow’s milk protein contain only more lysine (Table 2). Thus, the pancreatic enzymes used in the present study would more likely hydrolyze FIF and PIF than RIF. In any event, as with the gastric step of the digestion, the intestinal step exhibited significant differences between the three IFs when performed in dynamic conditions (Figure 2), while no significant differences were measured in static conditions. Moreover, the final DH were higher in dynamic conditions (around 50% up to 73%) than in static conditions (around 42% up to 52%) [34].

The difference observed between the static and the dynamic models can be explained by one-time versus continuous enzyme addition that might limit the enzymatic activity decrease, as a result of autolysis. One whole concentration (in the beaker) versus the progressive emptying of the products of the digestion might create inhibition of the digestive enzymes by some reaction products [71]. This phenomenon was also confirmed by Gauthier et al. (1982) [72], who suggested that the increased frequency of buffer replacement and increased emptying rate may reduce the inhibition of enzyme action by digestion products. This confirmed that within dynamic conditions, the enzyme inhibition should be lower than in static conditions. It further explained the higher DH values obtained after in vitro dynamic digestions. In any case, the dynamic model is closer to physiological conditions despite the same gastric emptying was applied for the three IFs while structural differences (Figure 1A,B) would suggest different emptying rates in vivo, as previously mentioned. Overall, the present study suggests a higher proteolysis for the two plant-based IFs, compared to the RIF, with the higher value obtained for FIF. Nevertheless, in vivo experiments are needed in order to confirm these in vitro observations.

### 3.4. Essential Amino Acid Bioaccessibility

During the course of the digestion in the intestinal compartment, the bioaccessibility of essential amino acids (EAA) progressively and significantly increased with mostly effects of IF and time separately (*p* < 0.001 or *p* < 0.05), whereas the IF × time effect was not significant (*p* > 0.05) (Table 4). This was the case for all EAA, except for isoleucine, where no significant difference was noticed between the three IFs and threonine, where a significant interaction between IF × time was noticed. In overall, RIF showed significantly higher bioaccessibility for tyrosine, phenylalanine, methionine, and cysteine compared to the two plant-based IFs. PIF was similar or slightly higher than RIF for lysine, leucine, and histidine, and was significantly higher for threonine than RIF and FIF. However, FIF showed the lowest values among the three IFs for lysine, leucine, cysteine, and histidine. The three IFs were equivalent for isoleucine and valine. Although comparisons between the three IFs are difficult because EAA bioaccessibility varied from one EAA to another, it suggests a higher quantity of free EAA for RIF compared to both plant-based IFs. This might be seen in contradiction with the higher DH measured for FIF and the lowest for RIF (Figure 2), but it should be kept in mind that the degree of hydrolysis is an indicator of all peptide bond cleavages, regardless if it results in free AA or peptide release, and regardless essential or non-essential AA are concerned. Lastly, both EAA bioaccessibility and DH should be considered jointly to better understand digestion.

In the previous study already mentioned above [34], the differences observed in terms of EAA bioaccessibility between RIF, PIF, and FIF at the end of intestinal digestion in static conditions were of the same magnitude as those observed in dynamic conditions. However, much higher values of EAA bioaccessibility were obtained in static conditions (from 12% to 88% for threonine and tyrosine, respectively) compared to the present study (from 7% to 54% for threonine and tyrosine, respectively). This difference might result from the continuous emptying while digestion occurs in dynamic conditions, whereas all the digestion products remain in the beaker in static conditions. Thus, more concentrated products and freer AA are expected in static than in dynamic conditions.

### 3.5. Size Distribution of Soluble Nitrogen Fraction and in Vitro Apparent Protein Digestibility of Infant Formulas

#### 3.5.1. Size Distribution of Soluble Nitrogen Fraction

Thanks to centrifugation, size exclusion chromatography and micro-Kjeldhal methods, the nitrogen fraction of in vitro digestas was categorized into seven classes according to solubility and molecular size (Figure 3). Around 15% of the total nitrogen was in the insoluble fraction, whereas around 80% corresponded to soluble peptides smaller than 10 kDa and free AA. Nitrogen fraction smaller than 10 kDa was considered as potentially absorbable form by the intestinal epithelium [73,74,75]. The majority of the nitrogen was in the 5–2 kDa class. There was no difference between the three IFs, except for FIF, which showed significantly more nitrogen in the soluble class higher than 10 kDa (0.5 point more) and less nitrogen in the free AA fraction (2 to 3 points less), in accordance with the slightly lower EAA release observed after digestion of FIF (Table 3). It should be noted that about 95% of the soluble nitrogen released from the three IFs was digested into either small peptides (< 10 kDa), which could potentially reduce the brush border enzymes into absorbable di or tri-peptides, or free AA that are directly absorbable by the intestinal epithelium [73,74,75].

#### 3.5.2. In Vitro Apparent Protein Digestibility

When in vitro digestion models are used to simulate the complex in vivo conditions of digestion, it is recognized that the apparent digestibility of protein can be estimated by calculating the percentage of the nitrogen present in the fraction of soluble peptides smaller than 10 kDa as suggested by other authors: Cave (1988) [73], Moughan et al. (1999) [74], and Huang et al. (2000) [75]. The latter used a dialysis tubing that retains molecular compounds of 12 kDa or more [75], which is close to the 10 kDa threshold chosen in our study.

In the present study, the apparent protein digestibility calculated was 74.9 ± 6.7, 89.2 ± 3.9, and 91.1 ± 3.1% for PIF, RIF, and FIF, respectively (Table 5). The limiting EAA are aromatic AA for RIF and isoleucine for PIF and FIF, with resulting PDCAAS-like scores of 67.5 ± 6.0, 75.4 ± 2.5, and 76.1 ± 3.3% for PIF, FIF, and RIF, respectively. PIF showed significantly lower in vitro digestibility compared to RIF and FIF and a lower PDCAAS-like score compared to RIF. The lower value obtained for PIF results mainly from a lower quantity of nitrogen recovered in the intestinal compartment and in the emptied fraction after 180 min of digestion compared to the nitrogen introduced into the digester. This may be associated to the large aggregates observed in the gastric compartment for PIF (Figure 1A) and that adsorbed on the wall of the digester (visual observation), resulting in a smaller quantity transferred into the intestinal bowl. In contrast, FIF was not significantly different from RIF for both in vitro digestibility and the PDCAAS-like score. Moreover, the EAA tryptophan could not be quantified with the method used for AA analysis (Section 2.4.3) but should be determined further in order to confirm the agreement of the IFs with the European regulation requirements [3]. Since tryptophan is often identified as a limiting AA in certain pulse cultivars, the PDCAAS-like scores should also be recalculated when the tryptophan content of the different IFs will be available.

Rudloff and Lonnerdal (1992) [76] studied the in vitro digestion of different IFs and determined a mean protein digestibility of 78.5 ± 4.2% for powder IFs (cow milk-based), a bit lower than our results for RIF and FIF, but closer to PIF values. Similarly, Nguyen et al. (2015) [28] reported in vitro digestibility of 81.5 ± 0.04% and 76.4 ± 0.04% for a standard cow’s milk-based IF and a soy-based IF, respectively. Ulloa et al. (1988) [32] tested chickpea protein concentrate as a potential milk substitute for follow-on IFs and measured an apparent protein digestibility (determined in vivo in growing rats) of 88.0% compared to 93.0% for a milk-based IF. Therefore, all these results are in agreement with the present results obtained for RIF and FIF, but somewhat higher than that for PIF, thereby confirming the high nutritional quality of faba bean proteins, and to some lower extent of pea proteins, with the aim of partially substituting whey proteins with plant proteins in IFs. Moreover, this also means that the proposed method to estimate protein digestibility using in vitro digestion coupled with size exclusion chromatography (with a threshold of 10 kDa) seems to be relevant as results are in accordance with the literature data including in vitro and in vivo results.

## 4. Conclusions

This study aimed to investigate the digestion of two plant-based IFs (PIF and FIF) compared to a dairy reference IF (RIF) using an in vitro dynamic model simulating infant digestion. In terms of microstructure, changes were observed only during gastric digestion with a faster aggregation for PIF and FIF, compared to RIF, likely due to specific sensitivity to pH drop. Very limited proteolysis was noticed during the gastric phase, whereas all three IFs were extensively hydrolyzed as soon as the chyme were submitted to the pancreatic proteases. Surprisingly, FIF showed higher DH value than RIF and PIF in the intestinal compartment after 3 h of digestion whereas no significant difference was observed in static conditions in a previous study (Le Roux et al., 2020) [34].

Finally, FIF and RIF showed similar in vitro apparent protein digestibility and PDCAAS-like score, higher than PIF. Thus, faba bean protein could be consider as a good candidate to partially replace whey protein in IFs without altering the nutritional quality and ensuring balanced nutrients intake compared to the infant needs. However, pea protein showed a lower protein digestibility than the reference under in vitro dynamic digestion and might likely need better dispersion to improve its nutritional properties. Nevertheless, the digestion model used in the present study, as well as the calculation chosen present some limits and need to be completed with in vivo studies in order to come closer to infant physiological conditions, and to confirm these promising results. These newly developed IFs should also complete economical and sensory satisfactions in order to be accepted by the consumer.

## Figures and Tables

**Figure 1 foods-09-00362-f001:**
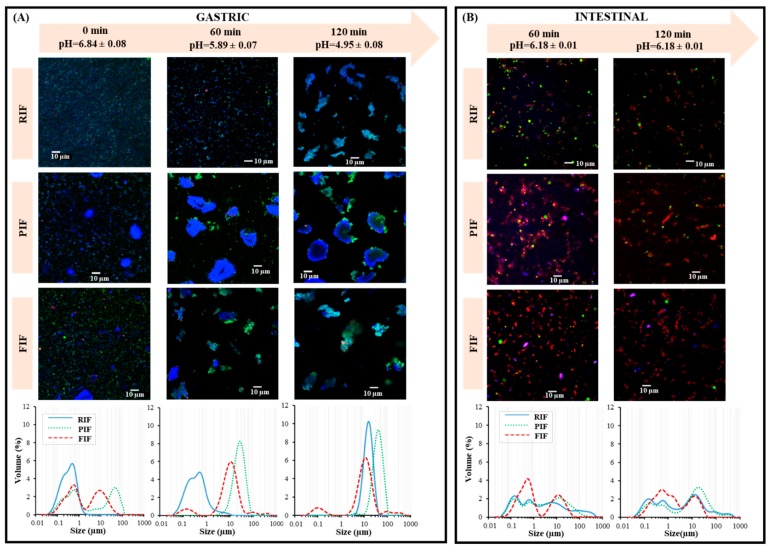
Confocal laser scanning microscopy images of RIF (reference infant formula), PIF (pea infant formula), and FIF (faba bean infant formula) in gastric (**A**) and intestinal (**B**) compartments over in vitro dynamic digestion and their corresponding particle size distribution profiles, as determined by laser light scattering. Proteins are colored in blue (FastGreen), apolar lipids in green (Lipidtox), and amphiphilic compounds in red (Rhodamine-PE).

**Figure 2 foods-09-00362-f002:**
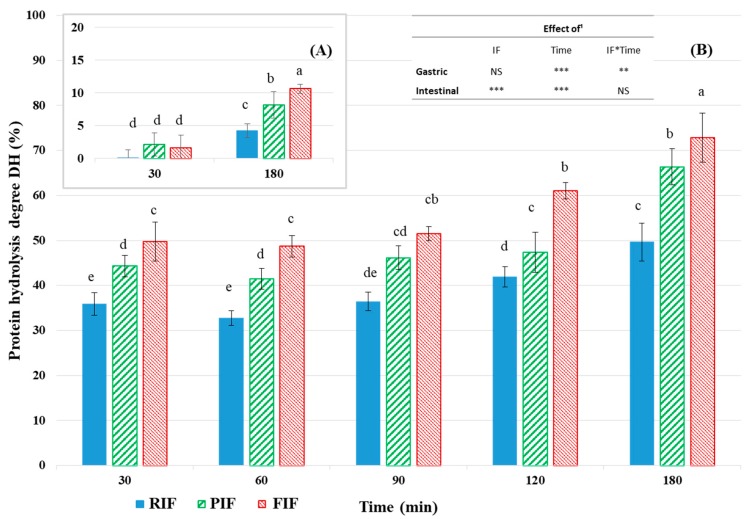
Degree of protein hydrolysis (DH) in gastric (**A**) and intestinal (**B**) compartments over in vitro dynamic digestion of RIF (reference infant formula), PIF (pea infant formula), and FIF (faba bean infant formula). Data are means ± SD (*n* = 3). Different superscript letters indicate significant differences (*p* < 0.05). ¹Statistics were completed on data from 30 to 180 min for the both compartments. The effects of IF, time and IF × time in the gastric and intestinal compartments are presented in the insert with statistical significance: *p* < 0.001 (***); *p* < 0.01 (**); *p* < 0.05 (*); and *p* < 0.1 (NS).

**Figure 3 foods-09-00362-f003:**
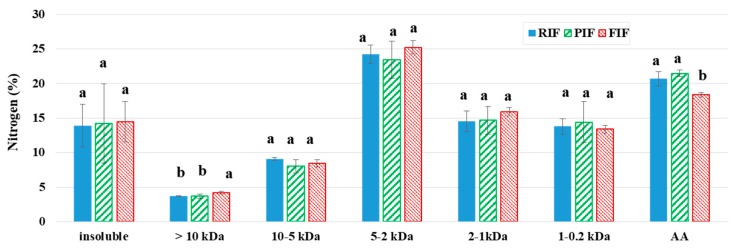
Nitrogen molecular size distribution (%) of RIF (reference IF), PIF (pea IF), and FIF (faba bean IF) determined at the end of in vitro dynamic digestion (in intestinal compartment and intestinal emptied fraction mixed). Data are means ± SD (*n* = 3). Values with a different superscript letter are significantly different (*p* < 0.05).

**Table 1 foods-09-00362-t001:** Gastrointestinal conditions for in vitro dynamic digestion. Adapted from de Oliveira et al. (2016) [42].

**Gastric Conditions (37 °C)**		
Simulated Gastric Fluid (SGF) (stock solution adjusted at pH 6.5)	Na^+^	94 mmol/L
	K^+^	13.2 mmol/L
	Cl^−^	122 mmol/L
Fasted state/initial conditions	SGF	2 mL
	pH	3.1
Infant formula ingested	Total volume	100 mL
	Flow rate	10 mL/min from 0 to 10 min
Gastric pH (acidification curve)	pH = −0.0155*t + infant formula pH with t: time after ingestion in min	
SGF + enzymes (pepsin)	Pepsin	268 U/mL
	Flow rate	1 mL/min from 0 to 10 min
		0.5 mL/min from 10 to 180 min
Gastric emptying [43,44]	t_1/2_ β	78 min 1.2
**Intestinal Conditions (37 °C)**		
Simulated Intestinal Fluid (SIF) (stock solution adjusted at pH 6.2)	Na^+^	164 mmol/L
	K^+^	10 mmol/L
	Ca^2+^	3 mmol/L
Intestinal pH	6.2	
SIF + bile	Bile salts	3.1 mmol/L
	Flow rate	0.5 mL/min from 0 to 180 min
SIF + pancreatin	Pancreatic lipase	90 U/mL
	Flow rate	0.25 mL/min from 1 to 180 min
Intestinal emptying [43]	t_1/2_ β	200 min 2.2

**Table 2 foods-09-00362-t002:** Nutritional composition of the three reconstituted infant formulas (IFs): RIF (reference infant formula), PIF (pea infant formula), and FIF (faba bean infant formula). Essential amino acid (EAA) composition is expressed in mg amino acid/100 mL infant formula. Data are means ± SD (*n* = 2). Different superscript letters indicate statistically significant difference (*p* < 0.05).

**Energy (kcal/100 mL)**	60.1 ± 2.1		
**Protein (g/100 mL)**	1.3 ± 0.03		
**Fat (g/100 mL)**	2.8 ± 0.1		
**Carbohydrates (g/100 mL)**	7.3 ± 0.3		
**EAA (mg/100 mL IF)**	**RIF**	**PIF**	**FIF**
Tyr	123.9 ± 0.4ᵃ	111.5 ± 3.4ᵇ	103.4 ± 0.8ᶜ
Lys	54.8 ± 1.9ᵇ	61.1 ± 3.5ᵃ	58.8 ± 1.2ᵃᵇ
Phe	54.5 ± 0.8ᵇ	68.4 ± 3.9ᵃ	60.9 ± 0.9ᵇ
Leu	145.2 ± 0.9ᵃ	124.5 ± 7.8ᵇ	118.0 ± 0.1ᵇ
Ile	77.4 ± 4.4ᵃ	65.4 ± 2.5ᵇ	60.1 ± 0.2ᵇ
Met	41.2 ± 5.3ᵃ	37.5 ± 0.7ᵃ	28.3 ± 0.7ᵇ
Cys	27.5 ± 0.2ᵃ	16.3 ± 0.3ᵇ	16.3 ± 0.1ᵇ
Val	83.9 ± 4.9ᵃ	76.2 ± 3.6ᵃᵇ	71.5 ± 0.8ᵇ
His	32.1 ± 1.5ᵇ	36.7 ± 0.5ᵃ	36.9 ± 1.3ᵃ
Thr	98.2 ± 1.6ᵃ	76.1 ± 3.0ᵇ	71.8 ± 1.1ᵇ

**Table 3 foods-09-00362-t003:** Mode diameters (µm) of RIF (reference infant formula), PIF (pea infant formula), and FIF (faba bean IF) samples in gastric (G) and intestinal (I) compartments over in vitro dynamic digestion.

Infant Formula	Compartment and Time	Mode 1 (µm)	Mode 2 (µm)	Mode 3 (µm)
RIF	G0	0.6		
	G60	0.6		
	G120	17.8		
	I60	0.2	1.1	5.6
	I120	0.2	0.6	15.8
PIF	G0	0.8	56.4	
	G60	31.7		
	G120	44.8		
	I60	0.3	1.2	15.8
	I120	0.2	0.8	17.8
FIF	G0	0.8	10.0	
	G60	0.2	12.6	
	G120	0.1	14.2	
	I60	0.6	11.2	
	I120	0.6	1.3	15.8

**Table 4 foods-09-00362-t004:** Bioaccessibility of essential amino acids (EAA) in the intestinal compartment (I) during in vitro dynamic digestion for RIF, PIF, and FIF. Bioaccessibility is expressed as % *w*/*w* of total amino acid. Data are means ± SD (*n* = 3).

EAA	IF	I30	I60	I90	I120	I180	Effect of ^1^		
							IF	Time	IF*Time
Lys	RIFᵃ	41.6 ± 1.3	46.2 ± 1.2	48.4 ± 3.9	55.0 ± 2.4	65.3 ± 3.4	***	***	NS
	PIFᵃᵇ	38.1 ± 4.7	41.7 ± 2.1	46.6 ± 1.3	49.4 ± 1.0	56.6 ± 3.4			
	FIFᵇ	33.6 ± 1.5	36.0 ± 2.1	42.3 ± 2.2	45.3 ± 2.1	54.0 ± 0.7			
Tyr	RIFᵃ	53.4 ± 1.1	55.0 ± 6.2	56.6 ± 7.5	54.3 ± 1.8	76.4 ± 3.3	***	***	NS
	PIFᵇ	42.5 ± 4.8	44.6 ± 1.6	50.0 ± 0.7	52.1 ± 0.6	58.2 ± 3.5			
	FIFᶜ	40.7 ± 0.6	43.4 ± 1.2	49.8 ± 1.9	51.5 ± 1.1	61.7 ± 0.7			
Phe	RIFᵃ	35.9 ± 3.0	42.4 ± 4.5	42.3 ± 3.5	48.1 ± 5.7	57.6 ± 3.8	***	***	NS
	PIFᵇ	24.8 ± 4.4	29.8 ± 2.9	34.9 ± 1.4	35.9 ± 1.1	42.9 ± 4.5			
	FIFᶜ	23.2 ± 1.1	27.8 ± 2.5	33.1 ± 0.7	36.0 ± 1.4	43.9 ± 1.9			
Leu	RIFᵃ	19.2 ± 0.2	23.6 ± 0.8	24.7 ± 3.1	29.4 ± 1.8	38.7 ± 2.2	***	***	NS
	PIFᵃᵇ	18.9 ± 3.0	22.0 ± 1.8	25.8 ± 1.5	28.4 ± 1.3	37.8 ± 2.9			
	FIFᵇ	13.4 ± 0.8	16.5 ± 1.4	21.0 ± 1.6	24.1 ± 1.6	33.2 ± 0.8			
Ile	RIFᵃ	6.2 ± 0.3	7.5 ± 0.4	7.9 ± 1.4	10.0 ± 0.9	14.8 ± 0.5	NS	***	NS
	PIFᵃ	6.4 ± 1.2	7.1 ± 0.6	8.8 ± 1.0	10.3 ± 0.8	18.1 ± 1.3			
	FIFᵃ	5.7 ± 0.1	6.3 ± 0.4	8.2 ± 0.6	9.7 ± 0.7	13.5 ± 0.5			
Met	RIFᵃ	16.9 ± 0.7	18.0 ± 1.9	19.1 ± 2.0	22.9 ± 1.4	26.5 ± 1.7	***	***	NS
	PIFᵇ	12.6 ± 1.2	12.8 ± 1.6	15.5 ± 1.0	16.6 ± 0.3	18.9 ± 0.3			
	FIFᶜ	11.5 ± 1.1	12.3 ± 0.7	14.7 ± 0.6	17.4 ± 1.4	21.1 ± 1.7			
Cys	RIFᵃ	22.3 ± 1.8	26.2 ± 5.2	27.7 ± 5.3	29.3 ± 0.9	18.1 ± 0.9	***	**	NS
	PIFᵇ	9.6 ± 5.2	14.5 ± 3.5	17.3 ± 4.3	15.9 ± 4.4	14.6 ± 2.4			
	FIFᶜ	4.7 ± 2.0	10.3 ± 4.8	9.4 ± 2.4	16.5 ± 5.6	12.2 ± 2.5			
Val	RIFᵃ	8.2 ± 0.2	10.4 ± 0.9	10.4 ± 1.6	13.3 ± 1.5	18.5 ± 0.1	*	***	NS
	PIFᵃ	8.5 ± 1.2	9.9 ± 0.3	11.4 ± 0.8	12.9 ± 0.6	19.4 ± 1.9			
	FIFᵃ	7.4 ± 0.2	8.0 ± 0.4	10.2 ± 0.5	11.5 ± 0.4	16.5 ± 0.3			
His	RIFᵃ	9.6 ± 0.3	7.6 ± 0.3	7.7 ± 0.7	9.6 ± 1.0	16.2 ± 1.2	*	***	NS
	PIFᵃᵇ	10.2 ± 3.5	7.3 ± 0.5	8.6 ± 0.8	9.5 ± 0.7	13.5 ± 2.3			
	FIFᵇ	7.2 ± 0.6	6.2 ± 0.6	7.0 ± 0.6	8.2 ± 0.1	10.4 ± 0.6			
Thr	RIFᵇ	6.8 ± 0.1ᵇ	6.7 ± 0.3ᵃ	6.4 ± 1.0ᵇ	7.5 ± 0.5ᵇ	11.0 ± 1.1ᵇ	***	***	*
	PIFᵃ	8.6 ± 0.6ᵃ	7.5 ± 0.4ᵃ	8.1 ± 0.6ᵃ	9.1 ± 0.9ᵃ	14.1 ± 1.3ᵃ			
	FIFᵃᵇ	7.9 ± 0.4ᵇ	6.6 ± 0.3ᵃ	7.4 ± 0.2ᵇ	8.3 ±0.2ᵃᵇ	10.8 ± 0.3ᵇ			

^1^ Statistics were completed on data from I30 to I180. Statistical significance: *p* < 0.001 (***); *p* < 0.01 (**); and *p* < 0.05 (*); NS (no significant differences). In case of a significant meal effect or of a significant interaction effect (IF*Time), multiple comparisons of means were conducted and differences between meals are indicated by a different superscript letter (*p* < 0.05), either in the IF column or at a given time point. Differences over time are not represented.

**Table 5 foods-09-00362-t005:** Limiting essential amino acid (LEAA), amino acid score (AAS), apparent in vitro digestibility (nitrogen fraction < 10 kDa), and PDCAAS-like score for RIF (reference infant formula), PIF (pea infant formula), and FIF (faba bean infant formula). Data are means ± SD (*n* = 3). Different superscript letters indicate significant differences (*p* < 0.05).

Infant Formulas	LEAA	AAS	Apparent in Vitro Digestibility (%)	PDCAAS-Like Score (%)
RIF	AAA^1^	0.85	89.2 ± 3.9 ᵃ	76.1 ± 3.3 ᵃ
PIF	Ile	0.90	74.9 ± 6.7 ᵇ	67.5 ± 6.0 ᵇ
FIF	Ile	0.83	91.1 ± 3.1 ᵃ	75.4 ± 2.5 ᵃᵇ

^1^ AAA corresponds to aromatic amino acids (phenylalanine, tyrosine, and tryptophan)

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
