# Peer review of "Are Faba Bean and Pea Proteins Potential Whey Protein Substitutes in Infant Formulas? An In Vitro Dynamic Digestion Approach"

_foods, 2020, doi:10.3390/foods9030362_

Round 1

Reviewer 1 Report

This is an interesting study, which may be important. As to the properties of fave bean proteins and pea proteins, a lot is known that is not described in the introduction that needs to be corrected. 

In the raw state, fava bean contains particular molecules such as phytic acid, saponins, lectins, alkaloids, and others, which can exert antinutritional functions reducing the digestibility of and leading to some pathologic conditions. That pea protein is almost resistant to pepsin digestion even at correct pH has also been described. So, please add these references and change the discussion after this. The proteins are encapsulated in lipid bodies, and in your system you do not have any lingual or gastric lipase, may be that could help. But the main problem is that the proteins are insoluble and precipitate and form large complexes that could not be hydrolyses. In the intestine there is no problem, but here you both add bile salts that act as detergents to split large particles and enzymes of high concentration that hydrolyse the mixture.

Please, add some words on the importance of drying method to get the powder.

It should also be discussed how presumptive infants will like to consume this powder.

Reviewer 2 Report

This manuscript presents novel data regarding the potential for faba bean and pea proteins as replacements for standard infant formulas.  This manuscript is well written and presents the case for use of faba bean protein as a substitute for whey proteins in infant formulas through identification of protein digestibility/availability and amino acid composition.  I do, however, have a few minor comments that should be considered by the authors.

Ln 52 – The authors present that Marinez et al 2000 as evidence for allergenic reactions in soybean.  This paper investigates chickpea protein, not soybean, and therefore may not be the optimal reference for this point. 

Ln 63 – I believe it would be beneficial to the reader if the in vitro protein digestibility of the IF’s discussed was presented explicitly. 

Ln 85 – A brief introduction to PDCAAS as a protein quality measurement would be appropriate to orient the reader to the important findings of this manuscript.

Ln 110-111 – What was the rationale for averaging the Jones factor for bovine milk proteins and pea/faba bean?  It would also be useful to mention that the factor of 5.4 is an average of legumes that had a range of 5.24 – 5.64 as presented in the Mariotti et al. 2008 reference. 

Ln 163-164 – It is unfortunate that the authors were unable to determine tryptophan content as that amino acid has been identified as the first limiting amino acid in certain pulse cultivars.  Lacking this information reduces the confidence in the PDCAAS-like score generated and should be mentioned in the appropriate discussion section.

Ln 220-227 – What was the authors rationale for choosing the reference pattern of the 2013 FAO/WHO guidelines rather than using the reference patterns for PDCAAS presented in the 1991 guidelines?

Ln 257-258 – Tyrosine was also significantly different between the plant-based IF’s according to Table 2.

Ln 266 – Table 2: The superscript for the FIF Thr content has both B and C.  Is this value significantly different from PIF as well as RIF?

Ln 403-427 Considering the research that has been done to demonstrate the bioaccesibility of EAA in the IF analyzed, it would be worthwhile to briefly discuss whether there is any value for certain amino acids to be more accessible to infants. 

Ln 509 – Table 5: I was surprised at Ile being identified as the limiting amino acid.  After checking the amino acid composition against the reference pattern in Table 5 of the FAO 2013 guidelines it appears that the Sulfur amino acids are first limiting with an AAS of 0.78 for PIF and 0.65 for RIF.  Having the sulfur amino acids being first limiting aligns with much of the previous literature on the protein quality of pulse crops.

Reviewer 3 Report

In general, the well-written manuscript is a solid work in developing a new infant formula product. Thoroughly enjoyed reading the whole article. It is highly recommended to publish. However, I observed some minor changes to be made (copied below) to increase the potential as well as the future use of the article.  

Line 46- Good techno-functionalities refer to a broad spectrum. Please be concise and a little explanation would provide clear information. E.g. is it in terms of processing or quality attributes after processing? I would suggest to include information from another paper (copied below) of the same author cited in ref 9.

Barać, M. B., Pešić, M. B., Stanojević, S. P., Kostić, A. Ž., & Čabrilo, S. B. (2015). Techno-functional properties of pea (Pisum sativum) protein isolates: A review. Acta periodica technologica, (46), 1-18.

Line 46- support the statement that the product would be of low cost

Line 70- Is there any IF in the market fortified with plant protein? If not, then the study would have huge potential as a new product development. If yes, please explain how the final product would make a difference. Information about this would increase the rationale of the study.

Line 106- Remove submitted, from the citation. The paper is already available online.

Line 110- Why the conversion nitrogen factor was 5.9 for PIF and FIF? Explanation and/or support from literature?

Line 522: Remove submitted, from the citation. The paper is already available online.

OVERALL: Is there/ could there be an impact of the non-protein element(s) of cow milk on the digestibility and bioaccessibility which could be different than the plant protein-based IFs?

Similarly, would this proposed plant-based protein impact on the storage stability of IF?
